# The Effects of Working Memory Training on Brain Activity

**DOI:** 10.3390/brainsci11020155

**Published:** 2021-01-25

**Authors:** Edward Nęcka, Aleksandra Gruszka, Adam Hampshire, Justyna Sarzyńska-Wawer, Andreea-Elena Anicai, Jarosław Orzechowski, Michał Nowak, Natalia Wójcik, Stefano Sandrone, Eyal Soreq

**Affiliations:** 1Faculty of Philosophy, Institute of Psychology, Jagiellonian University in Kraków, 31-007 Krakow, Poland; a.gruszka-gosiewska@uj.edu.pl (A.G.); michonowa@gmail.com (M.N.); natt.wojcik@gmail.com (N.W.); 2The C3NL Lab, Department of Brain Sciences, Faculty of Medicine, Imperial College London, London SW7 2BU, UK; adamdghampshire@googlemail.com (A.H.); andreea-elena.anicai@imperial.ac.uk (A.-E.A.); stefano.sandrone@imperial.ac.uk (S.S.); eyalsoreq@gmail.com (E.S.); 3Institute of Psychology, Polish Academy of Sciences, 00-090 Warsaw, Poland; jsarzynska@psych.pan.pl; 4Department of Cognitive Psychology and Psychology of Individual Differences, Wroclaw Faculty of Psychology, SWPS University of Social Sciences and Humanities, 53-238 Wrocław, Poland; jorzechowski@swps.edu.pl

**Keywords:** working memory, training, neural efficiency, N-back task, stop-signal task

## Abstract

This study aimed to investigate if two weeks of working memory (WM) training on a progressive N-back task can generate changes in the activity of the underlying WM neural network. Forty-six healthy volunteers (23 training and 23 controls) were asked to perform the N-back task during three fMRI scanning sessions: (1) before training, (2) after the half of training sessions, and (3) at the end. Between the scanning sessions, the experimental group underwent a 10-session training of working memory with the use of an adaptive version of the N-back task, while the control group did not train anything. The N-back task in the scanning sessions was relatively easy (*n* = 2) in order to ensure high accuracy and a lack of between-group differences at the behavioral level. Such training-induced differences in neural efficiency were expected. Behavioral analyses revealed improved performance of both groups on the N-back task. However, these improvements resulted from the test-retest effect, not the training outside scanner. Performance on the non-trained stop-signal task did not demonstrate any transfer effect. Imaging analysis showed changes in activation in several significant clusters, with overlapping regions of interest in the frontal and parietal lobes. However, patterns of between-session changes of activation did not show any effect of training. The only finding that can be linked with training consists in strengthening the correlation between task performance accuracy and activation of the parietal regions of the neural network subserving working memory (left superior parietal lobule and right supramarginal gyrus posterior). These results suggest that the effects of WM training consist in learning that, in order to ensure high accuracy in the criterion task, activation of the parietal regions implicated in working memory updating must rise.

## 1. Introduction

Imagine being immersed in a program that promises to train your brain so as to achieve augmented cognitive abilities. Some of the benefits you could expect are increased working memory capacity—the ability to remember information relevant for the task at hand irrespective of distractors—along with improved problem solving, reasoning, fluid intelligence, and emotional control. Is this an endeavor worth time, effort, and money? The “brain training” industry has seen a spectacular growth in the last decade with programs targeted for the wider public. Some companies promise real results for those who undergo regular training in the form of games, aimed to recruit and exercise working memory (WM) and other cognitive functions. However, whether or not their promises are actually based on empirical evidence has become one of the biggest controversies within the field of neuroscience [1,2,3,4,5]. The first controversy pertains to the question whether such training regimes really improve cognitive functions. The second one refers to the problem whether the observed improvements are measured at the neural level, thus justifying the term “brain training”.

### 1.1. Enhancement of Working Memory Capacity

Working memory is a psychological construct representing the cognitive function responsible for short-term storage of information and simultaneous manipulation of them [6,7]. Having replaced the formerly used notion of short-term memory [8], the concept of working memory combines the purely mnemonic functions with active processing of the temporarily stored information. It is therefore regarded to be at the core of cognition, contributing to virtually every act of higher-order information processing, such as thinking, reasoning, and problem solving [9], language processing [10], and language comprehension [11]. WM is thought to have several sub-systems and it is best understood as a framework for measuring and investigating cognition broadly, as opposed to a specific ability or brain network [12]. Its general efficiency, called working memory capacity (WMC) is regarded to be an individual trait: quite stable across various cognitive tasks but differentiated between individuals [13]. Multiple studies showed that WMC strongly predicts human intelligence, being its important cognitive substrate [14]. WMC also predicts school and academic achievement, as well as learning disabilities [15]. It is therefore not surprising that the neuroscience community have shown increasing interest in brain training, as it would provide a suitable method to improve cognition. Testing the validity of brain training is therefore not an ancillary, but a key objective.

Despite WM capacity being previously considered a stable characteristic of an individual [16], recent studies [17,18,19] have revealed that training on certain tasks can improve its functioning, on account of the brain’s plasticity. It is claimed that consistent and demanding engagement of these tasks would lead to a reinforcement of the WM neural circuits, subsequently leading to improved performance. Strengthening the structural connections in specific regions would facilitate other non-trained task that rely on the same structures, resulting in transfer effects [17]. Therefore, if training aimed to reinforce the WM network improved performance on an untrained task that relies on both WM and other networks, and that measures a different cognitive function, this would constitute evidence for transfer [18]. However, it is difficult to assess transfer effects, as improvements recorded for the untrained task might be specific for the task and not for the cognitive function it measures [19].

Research groups have focused on targeting specific functions which contribute to general cognitive abilities, with the hope of isolating specific neural networks that can be trained and improved. One such specific function, called working memory updating, is responsible for the quick replacement of stored items that are already irrelevant to the task at hand with more relevant ones. A laboratory task commonly used to assess WM updating is called N-back task. It consists in serial presentation of stimuli that are sometimes repeated at the given position, e.g., two (*n* = 2) or three (*n* = 3) elements after the first appearance. Increasing the *N* value, as well as speeding up the pace of stimuli presentation, makes the task more demanding. The N-back task is one of the most popular methods to assess WMC. It is also commonly used in many training programs. For example, Verhaeghen and colleagues [20] showed that 10 h of training on the N-back task results in significantly decreased response time, along with improved focus of attention. Similarly, Dahlin and co-workers [21] showed that by training participants on an updating task, they improve their performance in other tests that rely on updating, such as the 3-back WM task. Single-task training studies unanimously report improvements on the trained task, but evidence for transfer to other tasks is limited by them relying, to a certain degree, on overlapping brain regions and neural mechanisms. It is, however, unclear whether any or only the same operations of a brain network transfer across tasks [22].

Evidence for both transfer and its absence has been revealed in numerous papers, adding to the ongoing debate surrounding the effectiveness of cognitive training. The term ‘transfer’ can be interpreted as either improvement in: (1) operationally similar tasks, (2) different tasks measuring the same function, (3) different tasks measuring other functions, particularly fluid intelligence, or (4) everyday life, such as an increase in school achievement or reduction of unhealthy food consumption. Depending on the similarity between learning and testing situations, transfer effects are called near or far, although this is a continuum rather than dichotomy [23]. Some researchers strongly support the positive evidence for the potential of cognitive training to provide far transfer, including the increase of fluid intelligence. Perhaps the most compelling transfer evidence comes from Jaeggi and co-workers [24], who designed a dual N-back task and showed that training results in dose-dependent transfer to measures of fluid intelligence. However, a small sample size and the use of passive controls makes it likely for differential expectations to contribute to dissimilarities between the two groups [5]. Further research, summarized in a meta-analysis [25] allows for the conclusion that working memory training on the basis of N-back task brings about substantial improvement of general mental ability in healthy adults. Cognitive training leads to improved performance even in older adults, as Buschkuehl and co-workers [26] revealed in a cohort of 80-year-old adults who achieved increased memory performance on four different measures, compared to an active group.

There is also abundant evidence for an absence of transfer effects following WM training [27,28,29]. A very thorough analysis [5] highlighted that, while evidence for near transfer is less disputed, evidence for far transfer is practically non-existent, particularly if ecologically valid measures are adopted. A meta-analysis that included only the N-back training tasks [18] showed that ‘the only noteworthy transfer effect was seen to untrained N-back tasks’ (p. 1092). Moreover, a careful consideration of the limitations of many studies is essential before the utility of WM training becomes established. WM training studies do not commonly use active controls—participants that receive a training regime that is significantly less demanding and therefore elicits no effects—and instead use waiting-list or passive controls. However, a meta-analysis [30] shows that the overall effectiveness of training, expressed as the effect size, does not depend on whether the control group was active or passive. Also, using only one assessment task to demonstrate transfer gains may be problematic, as the improvement may be specific to the assessment task and not the broader cognitive ability it measures [31]. The general lack of consistency in the experimental study design of WM studies makes a comparison of results difficult.

### 1.2. Neural Effects of WM Training

Neural consequences of WM training are also controversial. On one hand, there are reports showing intensification of neural activity in regions involved in WM updating. For instance, Buschkuehl and colleagues [32] showed training-related increments in blood perfusion in the frontal and parietal regions implicated in N-back task performance. The training-induced increase of activation in multiple brain regions are also reported in the studies with children suffering from ADHD [33] and with adults who trained the dual N-back procedure in order to improve their regulation of emotions [34]. The review published by Constantinidis and Klingberg [35] suggests that neural activity in the prefrontal regions is increased due to WM training, not only in humans but also in other primates. Moreover, training tends to increase the strength of connectivity within the prefrontal regions, and between prefrontal and parietal regions as well [35].

On the other hand, there are studies reporting decline in brain activity following training [36,37,38]. For instance, Clark and co-workers [36] trained their participants with the dual N-back task and found that such a procedure decreased neural activation in the regions critical for WM functioning, with no transfer effects to fluid intelligence task. A study by Chang and co-workers [39] is particularly important because of the relatively large samples and follow-up measurement of WM training effectiveness. The authors found significant training related decrease of frontal activation in both HIV+ patients and healthy controls, which lasted for at least six months after the completion of training.

Still other authors [40] report differential changes in brain activity: critically important regions tend to get more activated after training, whereas other, relatively less important, regions tend to respond with reduced activation [41]. This pattern of training-induced neural changes is reported by Dahlin and co-workers [21]. Specifically, these authors found decreased activation in the frontal and parietal regions, which are implicated in WM activity, and decreased activation in the temporal and occipital lobes, as well as in subcortical areas (striatum). As to striatum, there are contradictory findings reporting increased activation of this structure following dual N-back training, with concurrent decreased activation of fronto-parietal areas [42]. Such a differential effect of training may reflect the phenomenon of neural efficiency. According to the efficiency hypothesis, the process of learning results in more efficient usage of brain resources, to the effect that the regions that are not critically important for task performance are less and less active as the task gets performed smoothly. It is also suggested that the initial phases of learning may be associated with increase of brain activation, whereas the later phases may show the opposite effect of decreased activation [32]. In general, the neural consequences of WM training are still not clear.

### 1.3. The Present Study

This study was motivated by the neural efficiency hypothesis [43,44]. According to this hypothesis, originally formulated in the research into the neural underpinnings of intelligence [45], highly able individuals demonstrate lower brain activation while doing a cognitive task, e.g., an intelligence test or a WM task. Later publications showed that such economizing on brain resources characterizes intelligent people only when the task at hand is relatively easy [46] or moderate in difficulty [47]. In more demanding tasks, intelligent people recruit even more brain structures than less intelligent ones, or recruit them more intensely [46]. Another important issue is the role of learning in the neural efficiency phenomenon. It has been suggested that the initial phases of skill acquisition require strong activation of relevant brain structures but in the course of learning these activations get weaker and weaker, or more focused [40,48].

Taking into account the above-mentioned evidence, we hypothesize that intensive WM training on the basis of the N-back task would result in relative lowering the level of brain activation in the neural structures subserving this task. By ‘relative lowering’ we mean that brain structures underlying task performance, though weakening with training, would be still activated above the baseline level. Next, we hypothesize that the degree to which respective structures would decrease their activation due to training would be moderated by individual differences, particularly the general mental ability (a.k.a. general fluid intelligence). In concordance with the previous studies, we expect that the efficiency effect would be particularly salient in the case of intelligent participants. Finally, we hypothesize that the relative changes of brain activation due to training (either lowering or heightening) would be moderated by the individual level of task accuracy. This expectation is based on the studies showing that differential recruitment of brain structures involved in task performance depends on the level of task performance [49].

In an attempt to verify these hypotheses, we devised a study in which participants underwent ten sessions of working memory training on the basis of the N-back task [50]. The training task was adaptive, that is, its difficulty level automatically adjusted to the individual level of performance. The same task was used during the scanning sessions, although in this case the difficulty level was moderate and constant both for the training group and the matched controls. We expected that the accuracy level of task performance in the scanner would be rather high and, importantly, not differentiated between the groups. An essential issue with verification of the efficiency hypothesis is levelling task’s demands. In order to asses if two groups of people differ in relative consumption of brain resources, they should not differ in accuracy. Only when both groups perform comparably well at the behavioral level, possible differences in activation of brain structures can be interpreted in terms of neural efficiency. Apart from the N-back task, participants were supposed to do another cognitive task that was not used in training. This task was introduced in order to check possible transfer effects. We chose a simplified version of the Stop-signal paradigm [51], which requires that an already activated behavioral tendency be contained in response to the signal of stop. Apart from allowing to assess possible transfer effects, this task also served as an estimation of the individual level of inhibitory control—a potential moderator of possible efficiency effects. There were three testing phases within the scanner: before training, between the fifth and the sixth training session, and after training. This decision was motivated by reports suggesting that the process of learning first leads to the increase of brain activation and only at its final stages may result in substantial decrements of activation. Three scanning sessions should allow to capture possible curvilinear effects of training. Finally, we applied the Raven’s matrices in order to assess the level of general fluid intelligence.

## 2. Materials and Methods

### 2.1. Participants

Forty-six healthy right-handed volunteers participated in the study. They were recruited via social media. First, we chose pairs of volunteering persons that were matched for age, sex and level of education and next the persons from each pair were randomly assigned to either the experimental or control group. The mean age of participants was 27.2 years (SD = 3.39). The female to male participants ratio was approximately 2:1. The mean level of education in both groups was 6.5 (an education level of 5 corresponds to secondary education, 6—bachelor’s degree or equivalent, 7—master’s degree or equivalent). Participants in the control group, who participated only in three fMRI scans, were paid 150 PLN (≅35 €), that is, 50 PLN/scan. Participants in the experimental group were paid 300 PLN (50 PLN/scan and 150 PLN for training). Originally, 50 people were recruited but two persons from the experimental group did not complete the whole training program; consequently, their counterparts from the control group were excluded, too. Ethical approval for the study was obtained from the Committee for Ethics of Scientific Research, Institute of Psychology, Jagiellonian University (Komisja ds. Etyki Badań Naukowych IPs UJ) on 29 May 2019 (the approval code: KE/04/052019). All subjects gave their written informed consent, including the information that they have the right to resign in any moment.

### 2.2. Study Design

The groups received fMRI scans in three distinct sessions. The experimental group underwent a cognitive training regime in between the three sessions. The participants performed two tasks in the scanner, namely the N-back task (which the experimental group trained on) and the stop-signal task (included to assess transfer), summarized in Figure 1. The participants were instructed and practiced the tasks before entering the scanner. The tasks were designed using Presentation (Version 18.0, Neurobehavioral Systems, Inc., Berkeley, CA, USA).

### 2.3. Tasks

#### 2.3.1. N-Back Task

The N-back task [50,52] required quick updating of working memory contents. The stimuli consisted of 17 geometric figures and patterns (Figure 1). Participants were presented with each stimulus for 1500 ms and a fixation cross was presented for 1000 ms before the next stimulus. Participants were instructed to press the button each time the figure presented on the screen matched the one presented N positions before in the sequence. The ‘targets’ presented on the wrong position (a.k.a. lures) were considered distractors. For instance, if the N number was set for 2, immediate repetitions (*n* = 1), as well as repetitions at the positions 3, 4 etc., should be ignored. Since the stimuli were constantly appearing and disappearing, participants had to update their mental sequence of events, because every new stimulus initially occupied the position *n* = 1, then the position *n* = 2, *n* = 3, and so on.

The N-back task was essentially the same in the training sessions outside the scanner and during three scanning sessions. An important difference pertained to the N number. In the scanning sessions, the N level of the task remained constant (*n* = 2). This decision was deliberate: we assumed that a relatively moderate difficulty level would result in lack of differences in accuracy between the groups. Looking for differences in brain activations, we decided to make the behavioral level of performance rather high and equal for both groups. In the scanning sessions, there were 128 stimuli presented in a sequence (run), including 16 targets, 15 lures (i.e., ‘targets’ repeated in a wrong position *n* = 1), and 97 distractors proper (i.e., not repeated stimuli). Each scanning session consisted of 3 runs. In the training session, on the other hand, the N number was adaptively changing according to an individual progress in performance. In other words, the difficulty level of the task, defined in terms of the N number, increased or decreased depending on the participants’ actual performance (see Section 2.4.1). Such a decision was motivated by the need to keep participants’ attention and effort at the highest possible level. The adaptive nature of training task is regarded the golden rule in the cognitive training research.

#### 2.3.2. Stop-Signal Task

The stop-signal paradigm [51] consists in quick responding to target stimuli. From time to time (usually in 20% of trials) the already activated response tendency must be terminated after an acoustic or visual signal of stop. The signal of stop occurs early after the target, which makes the task quite easy, or with increasing amount of delay, which makes the tasks rather demanding. An individual measure of inhibitory control is computed as the value of delay that ensures 50% of correctly inhibited responses. In order to asses this value, the time of delay is adaptively changing to a person’s performance. In this study, we simplified this paradigm in order to make it more convenient for an fMRI investigation. Specifically, we got rid of the adaptive nature of the task, instead providing the participants with two preset values of delay: short and long. The participants were presented with a sequence of stimuli—either X or O. They were asked to press as quickly as possible the appropriate buttons: LEFT if X is displayed on the screen and RIGHT if O is displayed on screen. However, occasionally, the sign ‘Do not press any button’ appeared in the frame. The delay for the presentation of the above-mentioned frame was either 0 or 200 ms. A correct inhibition of response was recorded as an ‘Inhibit-fast’ (following 0 ms message) or ‘Inhibit-slow’ (following 200 ms message) response. A failure to inhibit response was recorded as a ‘Fail-fast’ (following 0 ms message) or ‘Fail-slow’ (following 200 ms message) response. Each symbol was presented for 1500 ms and a fixation cross was presented for 1000 ms in between each stimulus. The run consisted of 200 stimuli. There were three runs in every scanning session.

The rationale for choosing the stop-signal task pertains to its being involved in one of the basic executive functions, that is, inhibitory control [51]. Together with working memory updating and switching, inhibition creates the commonly investigated set of executive functions [53]. Moreover, the N-back task used in the training, as well as during the scanning sessions, included lures that were supposed to be ignored. So, this version of the N-back task recruits not only working memory updating but also inhibitory control. We supposed that possible transfer effects would be more likely if two tasks would show some resemblance concerning their underlying cognitive mechanisms. There exist other inhibitory control tasks that could be chosen (e.g., go/no go or anti-saccades). We decided on stop-signal because of its being relatively well elaborated at the theoretical level [51].

#### 2.3.3. Paper-and-Pencil Instruments

We applied Raven’s Advanced Progressive Matrices, RAPM [54], in order to assess the general intelligence level. This part of investigation took place during a separate session, a few days before the first scanning session. We also used the Edinburgh Handedness Inventory [55] in order to ensure that all participants were right-handed.

### 2.4. Procedure

#### 2.4.1. The Training Procedure

The experimental group went through extensive training on a progressive version of the N-back task. Each training session consisted of 10 runs, and each run consisted of 40 stimuli. There were six targets (15% of stimuli) and six lures (15%) in every run. The performance level was defined as the proportion of the number of hits (i.e., properly detected signals) plus correct rejections to the overall number of trials in the given run. If such an index surpassed 90% in the given run, the difficulty level (i.e., the N number) in the next run increased. Conversely, if this index dropped down below the 75% level during two consecutive runs, the difficulty level in the next run decreased. Participants started to train every day beginning from the difficulty level at which they completed the last run in the preceding session. The training lasted for two weeks, five days per week. Each session lasted 30–35 min. Time of task depended on the length of short breaks in-between runs and individual reaction times (RTs), and remained stable across all 10 days of training.

#### 2.4.2. The Scanning Procedure

Whole-brain images were collected using a 3T GE Discovery MR750 scanner. During functional scans, a T2-weighted echo-planar image depicting blood oxygenation level-dependent (BOLD) contrast was acquired every 2.3 s with 3 × 3 × 3 mm resolution. Volumes consisted of a 64 × 64 × 41 matrix, 192 × 192 mm field of view, echo time (TE) = 30 ms, repetition time (TR) = 2.3 s & flip angle = 90°. The slicing acquisition order was automatically determined using GE software and classified as either ‘interleaved’ or ‘descending’. Across subjects, the N-back runs were approx. 201 TR’s in length (i.e., 462 s), and the stop-signal task was approx. 300 TRs in length (i.e., 690 s). EPI’s were collected in three independent sessions, each beginning with three N-back runs and ending with one stop-signal run. At the first session, an MPRAGE structural scan of 1^3 mm resolution was also collected for each participant with a 256 × 256 × 152 matrix, inversion time (TI) = 600 ms, TE = 3.68 ms, TR = 9.24 s & flip angle = 10°.

Pre-processing of the imaging data were performed using Statistical Parametric Mapping software (SPM12) and implemented in MATLAB 2018 (The Mathworks, Inc., Natick, MA, USA). Each EPI run was slice time corrected, (accounting for the acquisition order), followed by motion estimation and correction. After that, the mean EPI of each run was co-registered to the high-resolution structural scan. Then the structural volume was first segmented to different tissue types, and then spatially normalised to the Montreal Neurological Institute MNI152 stereotactic standard brain template according to the 12-parameter affine transformation and 16 nonlinear iterations. Using the T1 deformation matrix the realigned and unwarped functional volumes were normalised to the same standard space. Finally, functional volumes were smoothed using an 8^3 full-width half-maximum isotropic Gaussian kernel in order to reduce the inter-subject variability.

All functional sessions were modelled using SPM12 and the massive generalized linear model (GLM) approach. Where the association of each voxel, BOLD time-series and the experimental factors are independently modelled while controlling for realignment parameters as nuisance variables (i.e., adding the temporal displacement parameters estimated at the pre-processing stage as covariates in the model). For each session, we modelled all three runs from the N-back experiment together using the run constant as an implicit baseline, and the four N-back conditions (i.e., hits, misses, false alarms and no response) as experimental variables. At this level, three contrasts were defined: ‘Hits’, ‘Correct rejections’, and ‘Correct vs. Incorrect’. The first contrast (‘Hits’) represents differences in brain activations between correct identification of targets and all other types of responding (i.e., misses, false alarms, and correct rejection of distractors). The second contrast represents differences between the correct rejection of distractors and all other types of responding (i.e., hits, misses, ad false alarms). The third contrast refers to the difference between two types of correct responding (i.e., hits and correct rejections) versus two types of incorrect behaviour (i.e., misses and false alarms).

The stop-signal task was modelled independently, again using the constant as an implicit baseline and the six conditions (i.e., hits, misses, slow fails, fast fails, slow inhibition and fast inhibition) as experimental variables. Two contrasts were created: ‘Go vs. Stop’ and ‘Stop vs. Fail’. For both tasks (i.e., N-back and stop-signal) a 2nd group-level model was created, where contrast volumes from each group (trained vs. control) were collapsed to the session level (i.e., the within-session improvement was ignored) and the positive main effect of each of the contracts were examined.

#### 2.4.3. Data-Driven Region of Interest (ROI) Clustering

The resulting activation maps were cluster corrected using FDR (q < 0.05) and were segmented into independent ROIs [56] using an in-house implementation of the watershed transform [57] used to further investigate training effects.

## 3. Results

The effects of training, both behavioral and neural, will be analyzed in the ANOVA 2 × 3 factorial model with two independent variables. The between-subjects variable Group pertains to assignment of participants into training or control conditions. The within-subjects variable Session refers to three phases of fMRI scanning. According to the standard methodology of cognitive trainings [58], possible effects of intervention are usually expressed in terms of statistically significant interaction between Group and Session, to the effect that increments in the experimental group are expected to be more salient than those found in the control group. The latter may also improve their performance due to repeated measurement. The effects of individual differences will be analyzed in the GLM model with individual dimensions treated as continuous covariates. Also, contrast estimates will be correlated with individual dimensions across participants.

### 3.1. Behavioral Results

First, we checked if the training group improved their N-back task performance across ten training sessions. The number of hits per day during the second phase of training, consisting of sessions 6–10, was significantly higher than the number of hits per day during the first phase of training, including the sessions 1–5 (ANOVA repeated measures, M_1_ = 754.87, SD = 246.82, M_2_ = 1013.17, SD = 351.47, *F*{1,22} = 30.44, *p* < 0.001, partialled *η*^2^ = 0.581). Since the task was adaptive, adjusting its difficulty level according to every participant’s progress, the average N number during the two phases of training was also examined. It appeared that the average N level was significantly higher during the second stage of the training in comparison to the first stage (M_1_ = 3.95, SD = 0.91, M_2_ = 5.53, SD = 1.85, *F*{1,22} = 32.51, *p* < 0.001, partialled *η*^2^ = 0.596). These findings, particularly the values of size effect, symbolize a huge improvement of the N-back task performance across ten training sessions. In order to check whether the progress took place in both phases of the training, we computed the difference measures by subtracting the number of hits in the preceding session form the number of hits in the following session (e.g., 2−1, 3−2, and so on). We found that these difference measures were significantly greater than zero in both the first and the second phase of the training (*t*{1,22} = 7.37, *p* < 0.001, and *t*{1,22} = 9.03, *p* < 0.001, respectively). These findings justify a conclusion that the training was effective in terms of heightening the level of performance in the trained task. The so-called mere practice effects are not particularly important per se, but they imply that the training regime was effective, and the study was internally valid.

Having checked for the mere practice effects, we analyzed the near transfer effects. For that, we analyzed the results of the same N-back task performed in the scanner, and the results of the non-trained Stop-signal task, also performed in the scanner. The general linear model with 2× group (between-subjects) and 3× scanning sessions (within-subjects) revealed that, if the number of hits was regarded, only the effect of session was statistically significant (*F*{2,88) = 22.20, *p* < 0.001, partialled *η*^2^ = 0.335). The observed increase took place only between the first and second session; the third scanning sessions brought about similar results as the second one. As to the number of correct rejections, again only the effect of session appeared significant (*F*{2,88) = 12.41, *p* < 0.001, partialled *η*^2^ = 0.220) and the analysis of contrasts revealed that the progress took part not only between session one and two (*p* < 0.001) but also between session two and three (*p* = 0.035). The effect of group and the interaction effect were not significant. Basic descriptive statistics for these analyses are included in Table 1. Generally, the findings suggest lack of any training effects; such effects could be claimed only if the group by session interaction would obtain statistical significance. Since both groups increased their performance in consecutive sessions in a similar pattern, one can speak only about the effect of repeated measurement (test-retest).

The same conclusion can be drawn on the basis of the Stop-signal task. Its easy version (zero-delay of stop) produced a ceiling effect: most participants did not commit any error and only some committed 1–3 false alarms. Therefore, data from this version were not subjected to inferential statistics. As to the hard version (signal of stop delayed 200 ms), there was enough variance to perform inferential analyses: the great mean was 23.78 of correct inhibitions per run consisting of 31 such events (SD = 4.99, range 11.33–30.33). Still, performance in the hard version of the Stop-signal task did not depend on group, neither on the group by session interaction. Even the main effect of the session was insignificant, suggesting a lack of test-retest impacts.

### 3.2. Imaging Results

The cluster maps provide focused ROIs suitable for analyzing the main effect of change across sessions and the session × group interaction. In the ‘Hits’ contrast (Figure 2A), the change in activation across sessions in the two groups is focused predominantly on regions of the frontal and parietal lobes. Three clusters of activation appeared significant after the FDR correction: one frontal (superior frontal gyrus) and two parietal (superior parietal lobule and supramarginal gyrus posterior). The coordinates of the overlapping ROIs, together with their anatomical labels and corresponding Brodmann’s area numbers are shown in Table 2. Although these activations are strong enough and confirm the well-established findings concerning the neural substrate of working memory [52,59,60], the peak voxel mean activation of the significant clusters shows no significant effect of training. In the ROI 1 (superior parietal lobule) activation initially increased between the first two sessions and then it dropped down between sessions two and three, but this is only a trend (*F*{2,88) = 2.54, *p* < 0.085, partialled *η*^2^ = 0.056). Most importantly, a lack of interaction with the group indicates that these changes were due to repeated measurement rather than training. ROI 2 (supramarginal gyrus posterior) did not show any significant effect. As to the ROI 3 (superior frontal gyrus), a very weak effect of session (*F*{2,88) = 2.69, *p* < 0.073, partialled *η*^2^ = 0.059) indicates that activation in this region was weaker and weaker with consecutive scanning sessions. Lack of interaction with the group factor, however, suggests that this tendency resulted from repeated measurement, not from training outside the scanner. Altogether, these findings clearly demonstrate that, as far as the ‘Hits’ contrast is regarded, no neural effects of cognitive training were observed.

In the ‘Correct rejection’ contrast, the change of activation was also observed in the frontal and parietal regions. Specifically, three clusters of activation remained significant after the FDR correction (Figure 2B): paracingulate gyrus, superior parietal lobule, and supramarginal gyrus posterior (see Table 2). The examination of session, group, and the interaction of both factors, showed lack of any statistically significant effects. These results pertain to all clusters of activation mentioned above. Again, a preliminary conclusion may be drawn that no neural effects of training appeared at all, as far as the ‘Correct rejection’ contrast is concerned.

Exactly the same clusters of activation appeared significant for the contrast ‘Correct vs. Incorrect’ (Figure 2C). ROI 1 (superior parietal lobule) showed a curvilinear pattern of change: after the increase between the first and the second sessions the activation dropped down in the third session, but this is only a trend (*F*{2,88) = 2.72, *p* < 0.071, partialled *η*^2^ = 0.060). Again, a lack of any significant interaction with the between-subjects factor of the group suggests that these changes resulted from repeated measurements, not training. Of course, three consecutive measurement sessions also provided learning opportunity but this is not was we hypothesized. ROI 2 (supramarginal gyrus posterior) did not show any systematic relationship with the factors of session, group, or interaction between both. As to the ROI 3 (paracingulate gyrus, Table 2) we found a significant decrement of activation between sessions (*F*{2,88) = 4.12, *p* < 0.020, partialled *η*^2^ = 0.087). It appeared that the activation in this region dropped down between the first and the second session, as well as between the second and the third. Thus, a trace of neural efficiency occurred. However, lack of the session x group interaction suggests that this trend depended on repeated measurement rather than cognitive training.

As to the stop-signal task, changes in activation concerning the ‘Stop vs. go’ contrasts can be observed mainly in the inferior frontal cortex and pre-supplementary motor areas (Table 3), confirming previous reports [61,62,63]. The peak voxel mean activation of the significant clusters shows no significant differences between the training and control groups across the three sessions. Non-significant trends of increased activation can be observed in the training group, while changes in activation are less evident in the control group. The significant clusters of activation associated with the ‘Stop vs. fail’ contrasts are focused in cortical motor areas. No significant trends can be observed across groups or sessions. The results of the second-level full-factorial analysis for the significant clusters of activation in the stop-signal task support the conclusion about lack of any transfer effects of training observed at the neural level of analysis.

### 3.3. Individual Differences

The analyses reported in Section 3.2 are based on the ANOVA factorial model with one within-subjects variable (session) and one between-subject factor (group). The results clearly demonstrate that the first hypothesis (training-induced neural efficiency) did not obtain empirical support. However, lack of general effects of efficiency does not exclude a possibility that interesting trends of activation change may be connected with individual variables. We took into account three such variables: (1) general mental ability, measured with Raven’s matrices, (2) inhibitory control, measured with Stop-signal task, and (3) WM updating, measured with N-back task accuracy. Variables (2) and (3) have been computed as the average scores across three scanning sessions. N-back accuracy was operationalized in two ways: as the number of hits and the number of correct rejection of lures. In order to check for possible moderation caused by these variables, we included every one of them into the general linear model as continuous co-variables.

In the first contrast (‘Hits’), we did not find any significance of intelligence and inhibitory control for relative change of brain activation. An interesting and quite strong effect of N-back accuracy, measured with the number of correct rejection of lures, appeared in the left superior parietal lobule (ROI 1, Table 2): the more accuracy the higher the activation in this region (*F*{1,43) = 11.71, *p* < 0.001, partialled *η*^2^ = 0.214). The accuracy variable also interacted with session (*F*{2,86) = 3.25, *p* < 0.044, partialled *η*^2^ = 0.070): the less accurate participants showed a tendency to reduce the activation in this region, whereas the more accurate ones kept this region relatively active, particularly in the second and third session (see: Figure 3). In the right supramarginal gyrus posterior (ROI 2, Table 2), the effect of accuracy was quite similar (*F*{1,43) = 6.03, *p* < 0.018, partialled *η*^2^ = 0.123). However, the interaction effect was insignificant.

In the second contrast (‘correct rejections’) we found a significant interaction between inhibitory control, measured with the number of correctly inhibited responses in the Stop-signal task (hard version), and the factor session (*F*{2,86) = 4.90, *p* < 0.010, partialled *η*^2^ = 0.102). This effect pertained to the ROI 2 (paracingulate gyrus). It appeared that people scoring low in the stop-signal task reduced activation in this region in the final scanning session, whereas those who performed relatively better increased their activation in this session. No further effects have been observed in this cluster of activation. As regards two other clusters (superior parietal lobule and supramarginal gyrus posterior), we did not find any relationships between relative change of activation and individual differences.

As to the third contrast and ROI 3 (paracingulate gyrus), we found an interesting interaction between inhibitory control, measured with the number of correctly inhibited responses in the difficult version of the Stop-signal task (200 ms delay), and session (*F*{2,88) = 3.72, *p* < 0.028, partialled *η*^2^ = 0.078). It appeared that less accurate participants lowered their activation in the third session down (contrast estimates not differing from zero, whereas more accurate participants kept their activation at the level significantly higher level. This result suggests that it is good for inhibitory control to keep the paracingulate gyrus active enough even in the third approach to the stop-signal task. Those who responded with lowering its activity earned fewer number of correctly inhibited responses. This interaction is the only effect we found in the frontal region. In particular, intelligence did not predict anything. In the parietal regions, we found that only N-back task accuracy predicted relative changes in brain activation. Specifically, accuracy operationalized as the number of correct rejection of lures predicted brain activation in the ROI 1 (superior parietal lobule): (*F*{1,43) = 7.52, *p* < 0.009, partialled *η*^2^ = 0.149); in the ROI 2 (supramarginal gyrus posterior) the significance of this effect seems problematic (*F*{1,43) = 3.50, *p* < 0.068, partialled *η*^2^ = 0.075). Anyway, these relationships show the increase of brain activation with rising level of task accuracy. Moreover, an interaction appeared significant (*F*{2,86) = 3.66, *p* < 0.030, partialled *η*^2^ = 0.078), to the effect that less accurate participants responded with lower and lower activation across consecutive sessions in superior parietal lobule (ROI 1), whereas more accurate participants increased their activation in this region in the second session and kept it at this heightened level in the third session (Figure 4). In ROI 2, this interaction was insignificant. Importantly, the effects reported in this paragraph occurred only if lure rejection was regarded the index of task accuracy; if the number of hits was taken into account, they did not cross the level of statistical significance. Also, neither intelligence nor inhibitory control brought about any significant results.

Analyses performed at the group level may suggest lack of any training-related effects. However, brain activations in the ROIs that survived the false discovery rate (FDR) correction depended on individual characteristics of participants, particularly on the N-back accuracy measured with the number of correct rejection of lures. The question arises if accuracy is linked with brain activations not only across sessions, as visualized in Figure 3 and Figure 4, but also within sessions. To check this possibility, we conducted a series of correlational analyses, in which individual levels of brain activation were regressed on chosen individual variables. Only the significant ROIs were taken into account (see Table 2). Since most of the variables did not conform with the normal distribution, we used Spearman’s *r* coefficient.

First, we checked the correlations between two measures of N-back task performance, the number of hits and the number of correct rejection of lures (i.e., targets in the “wrong” position) with relevant brain activations. The time of performance was matched, that is, we correlated the number of hits and correct rejections in the first session with activations only in the first session, and so on. We found that the number of hits did not enter in any correlation with strength of activation in the ROIs that are included in Table 2. The number of correct rejection of lures, on the other hand, correlated positively and significantly with strength of activation in ROIs 2 and 3 (contrast ‘Hits’, see Figure 2), and also with activation in ROIs 1 and 2 (contrast “Correct vs. Incorrect’, see Figure 2). However, these relationships took place only in the third scanning session, that is, after training. After splitting the data, these correlations disappeared in the control group but became even stronger in those who underwent training. The results referring to the third contrast are included in Table 4.

Interestingly, these two parietal regions (left superior parietal lobule and right supramarginal gyrus posterior) changed systematically with training session as far as their general activation level is concerned. Although these patterns of change did not show any symptoms of being affected by intensive N-back training outside scanner, they suggest the influence of repeated measurement on activation of parietal parts of the network subserving N-back task performance. Now we can see that the effect of training may amount to the positive correlation between the number of correct rejection of lures and the activation change. It seems as if the training group learned how to ensure successful rejection of lures through increasing brain activation in the parietal parts of the frontal-parietal network that subserves the functioning of working memory. The frontal part of this network did not show such a pattern of change following training.

## 4. Discussion

We investigated the neural consequences of working memory training. Young healthy volunteers underwent the training regime consisting of ten sessions during which they practiced an adaptive version of the N-back task. The same task but in its nonadaptive version was also used in three scanning sessions that took place before training, after five training sessions, and after termination of training. Randomly assigned control volunteers were also scanned three times but they did not undergo any organized training in between of scanning sessions. The trained task’s performance improved substantially as an effect of training but no transfer effects in the untrained stop-signal task’ scores have been found. We were also unable to identify any change of activation in the task-related brain clusters, which could be linked to training outside the scanner. Such changes appeared, however, as a consequence of mere repetition of measurement within scanner. We found that the activation in the frontal areas subserving N- back task performance (superior frontal gyrus) tended to decrease their activation linearly between the first and the last session, whereas the parietal areas (left superior parietal lobule) showed a curvilinear relationship, first increasing its activation and next becoming relatively less active. We also found that activation in the parietal regions depended on task accuracy: participants who obtained high level of accuracy, measured with the number of correct rejections of luring distractors, responded with increased activation in left superior parietal lobule and right supramarginal gyrus posterior, and those with lower accuracy indices responded with decreased activation. This tendency was particularly visible in the second and third scanning session. General mental ability (intelligence) did not show any moderation on task-related brain activity. We also found that activation in the above-mentioned parietal regions was positively correlated with accurate rejection of luring distractors. However, these relationships were evident only on the third scanning session and were substantially stronger in the training group, in comparison to the controls.

First of all, we suggest excluding the possibility that the training regime was not effective enough. We found very large scaled practice effects that justify the conclusion that the training group really improved their N-back scores during training. These improvements took place throughout the whole training program, meaning that they were not limited to initial phases of the process. It seems that the training procedure was effective and the experimental manipulation was successful, hence the whole study was internally valid. Therefore, a lack of training-induced effects measured both at the behavioral and neural levels of analysis, as well as lack of any symptom of transfer to the untrained stop-signal task, should be regarded as an argument against the position claiming the existence of significant ‘brain training’ effects. Thus, we have to reject the first hypothesis that predicted training-related decrease of neural activation in the regions involved in the N-back task performance. This conclusion is consistent with some findings [64], although it is inconsistent with others [32,36,37,38].

Another possibility that we would like to exclude pertains to the supposedly low difficulty level of the tasks that the participants performed while being scanned. It has been found in the research on intelligence [47] that neural efficiency occurs if a task at hand is moderate in difficulty. If it is too easy, everybody performs correctly without spending much effort and neural resources. If it is very demanding, even the most able individuals have to perform with the highest possible engagement. Only if the task is moderately difficult are those who are comparatively more endowed able to work with a relatively lesser amount of engaged brain resources. In our study, the level of intelligence did not yield any significant results, but a similar line of reasoning may be applied in reference to skill acquisition. One can speculate that only the moderately difficult task allows for differentiation in recruitment of relevant brain clusters between those who train and those who do not. If the task is too easy or too difficult, one should not expect training-induced effects of neural efficiency. It seems, though, that our scanning task was moderate in difficulty. Notwithstanding certain symptoms of the ceiling effect, there was enough room for accuracy improvement between the scanning sessions (Table 1). We deliberately set the N number at the relatively low level (*n* = 2) expecting lack of between-group differences in accuracy (indeed, they did not occur) and thus being able to concentrate on possible neural effects of training. However, the *n* = 2 level still ensured enough variance to assess between-session differences in accuracy. In our opinion, only the *n* = 1 level would make the task trivial.

The second hypothesis did not obtain empirical support, either. The level of intelligence, assessed with the advanced version of Raven’s matrices, did not enter in any relationship with task-related changes in brain activation. The same finding pertains to inhibitory control, measured with accuracy in the harder version of the stop-signal task (200 ms delay between target and stop). Investigation of inhibitory control was exploratory, since there is no evidence in the literature that this dimension of individual differences has been investigated in the context of the neural efficiency hypothesis. Moreover, the stop-signal paradigm is not a test, but an experimental procedure devised to investigate cognitive control processes. Although it has been used in some psychometric investigations, its properties as a test are dubious. Therefore, these preliminary results should be treated with caution. Raven’s matrices, on the other hand, are well-established and reliable instrument to assess human general mental ability. A lack of any significant symptoms of neural efficiency connected to intelligence may be accounted for in terms of differences in methodology. The neural efficiency hypothesis in intelligence has been verified mostly in studies using PET [45] or EEG [44,47]. Maybe the efficiency phenomenon is difficult to capture with the fMRI methodology. Another probable explanation, pertaining to possible attenuation of intelligence scores, seems not viable because our sample did not differ from the general population in Raven’s scores. However, the small sample size may be responsible for lack of the neural efficiency symptoms related to intelligence.

As to the third hypothesis, we found that accuracy with which the participants performed the N-back task predicted brain activation in the left superior parietal lobule. The main effect shows that heightened levels of accuracy are associated with increased activation in this region. The interaction effect, visualized in Figure 3 and Figure 4, amounts to decreasing activation in the left SPL with consecutive scanning sessions by less accurate people; the more accurate ones increased the left SPL activation in the second and third sessions. However, these effects were not modulated by the group factor, meaning that participation in the N-back training outside scanner did not contribute anything. Therefore, the third hypothesis should also be excluded. We hypothesized that task accuracy would moderate the training-induced effects of efficiency, but such effects are lacking. However, three consecutive scanning sessions also provide an opportunity to train the N-back task. This is not the training we expected to occur but it seems fairly possible that the participants learned how to deal with the task taking the opportunity to do it three times with one-week break in between the sessions. If so, the effect shown in Figure 3 and Figure 4 may reflect the joint effect of the process of learning and task accuracy. Some participants might have learned how to improve their performance thanks to the increased activity in the left parietal region. Others did not improve their performance, probably due to decreasing activation in this region. Why the training proper, taking place outside the scanner and much more intensive in terms of intensity and learning time, did not bring about similar results remains problematic. Perhaps the training gains were limited to the mere practice effects; potential improvement in the same task but performed in different conditions should be interpreted in terms of near-transfer effects, that did not emerge. Although near transfer is usually defined as potential improvement in a different task engaging the same cognitive function, it seems that a task identical with the trained one but performed in different conditions (i.e., within scanner) should be regarded different enough to justify the term ‘transfer’. Still, even this kind of ‘nearest transfer’ did not occur in our study.

Unexpectedly, we found that activation in the parietal regions were positively correlated with task performance accuracy, measured with the number of correct rejection of lures. So, the impact of accuracy on brain activation in selected regions occurred not only between subjects (low vs. high accuracy) and between scanning sessions, as can be seen in Figure 3 and Figure 4, but also within subjects. In other words, accuracy generally increased these activations, particularly in the last two sessions, but also predicted the selected activations on the individual level. These two tendencies probably work independently, as it is theoretically possible to find such correlations without the main effect or vice versa. Importantly, these correlational relationships occurred only in the third scanning session and were substantially stronger in the training group in comparison to the controls. It seems as if the training group learned how to ensure high level of accuracy through heightening the level of activation in left superior parietal lobule and right supramarginal gyrus posterior. In fact, this is the only training-related effect that we were able to find, as far as the training outside scanner is concerned. To our knowledge, such effects have not been reported so far. Their interpretation is therefore quite difficult, the more so that other measures of accuracy, i.e., the number of hits, did not show such a relationship. The version of N-back task that we have used is a bit dualistic nature: it needs quick reaction to targets and efficient inhibition of non-targets. Only the second component brought about significant correlational effects. Moreover, these effects did not occur in the frontal regions involved in the task performance. Perhaps further research will help to confirm these findings and suggest suitable interpretation.

We were unable to find empirical support for the neural efficiency hypothesis, according to which the process of learning causes decreased activation of brain structures implicated in cognitive task performance [43,44,45,46]. Maybe neural efficiency is an empty notion and the relevant empirical findings can be interpreted alternatively [65]. It seems that, instead of general lowering brain activations following learning, one should expect differential weakening and strengthening of brain activations. Such differential changes may result either from the process of learning (such as WM training) or from individual differences. Our results suggest the second possibility.

Conclusions resulting from our study are limited in several ways. Firstly, these conclusions are restricted to working memory training based on the N-back task. Research with other tasks may bring about more positive results, although it is worth to underscore that contemporarily the N-back task epitomizes the function of working memory updating, which is crucial for WM efficiency. Also, more complex tasks, particularly commercial games, may show increased effectiveness as a mean to train cognitive functions. Being rather complex, and engaging wide variety of cognitive processes, commercial games are probably more advisable from the practical point of view. However, their complexity precludes precise assessment what kind of mental process makes a cognitive training work, if at all. Secondly, our conclusions refer to the brain functions rather than structures. There are studies reporting that working memory training improves white matter integrity in task-related cortical areas and in the corpus callosum [66,67]. Similar effects have been observed in the investigation of neural consequences of playing commercial real-time strategy games [68]. Functional connectivity has also been demonstrated to change after working memory training [66]. Thirdly, these conclusions pertain to healthy young volunteers. Such a sample was deliberately recruited for this study in order to eliminate possible contribution from the side of aging or behavioral disorders. Studies with senior citizens or children, particularly with developmental disorders such as ADHD, have already demonstrated that working memory training can affect the functional aspects of neural organization [32,33].

One can also raise criticism for duration of training that our participants underwent. Although their training regime was quite intensive and comparable to procedures adopted in other studies, it was nevertheless rather short-term. Hampshire and co-workers [1] found evidence of near transfer after very long time of brain training (months or years). It may be the case that much longer-term training is required to produce substantial scaled transfer, even to the most similar tasks. Another caveat pertains to the fact that the control group was not active. Making the controls train on another task, possibly not very demanding for working memory, could allow for stronger conclusions. Still better solution would be to recruit two control groups: one passive and one active, as is the case in some training studies [27]. However, this issue may not be that important as the meta-analysis published by Karbach and Verheaghen [30] showed that, in working memory trainings, the active and passive control conditions give ‘indistinguishable’ results. Finally, the sample size is an issue. Although many other neuroimaging studies on WM training are based on comparable or slightly bigger samples [33,35,41,66], forty-six participants is a number that may put into question the conclusions concerning individual differences.

## 5. Conclusions

The key finding of this study amounts to the conclusion that accuracy in the N-back task performance is associated with a bilateral increase of activity in parietal regions involved in task performance. The relationships between accuracy and relative level of activation in the left superior parietal lobule and right supramarginal gyrus posterior occurred in both between subject comparisons (high vs. low accuracy people) and within subject correlational analyses. Moreover, N-back training resulted in an increased positive correlation between accuracy and activation in these parietal regions. We conclude that the training group learned how to improve their task performance through increasing brain activation in the selected parts of the parietal cortex.

## Figures and Tables

**Figure 1 brainsci-11-00155-f001:**
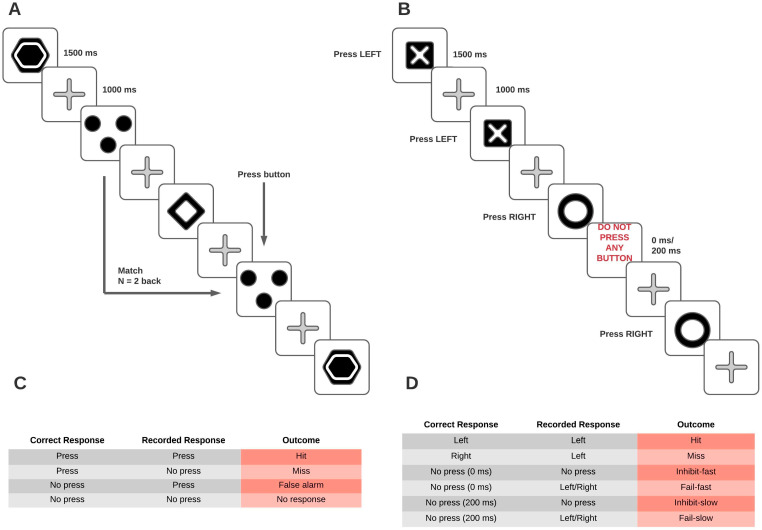
Schematic representation of the tasks used in the three scanning sessions. (**A**) shows an example of the stimuli the subject would see when performing the N-back task. (**B**) shows an example of the stimuli the subject would see when performing the Stop-Signal task. (**C,D**) respectively show the variables recorded for each run of the N-back and Stop-signal tasks (‘Outcome’ column), while also offering an explanation for when a response was considered correct.

**Figure 2 brainsci-11-00155-f002:**
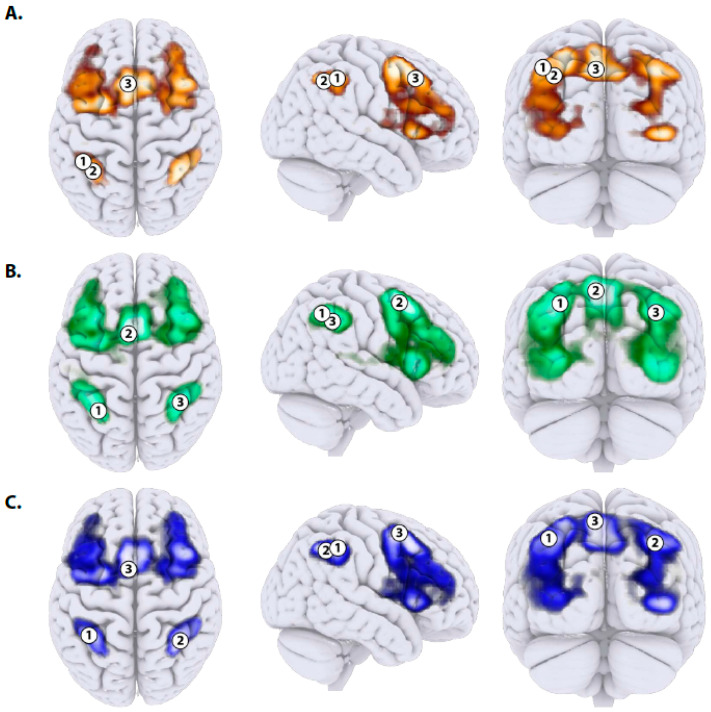
Contrast maps showing changes in brain activation associated with the correct identification of targets (‘Hits’, 3A), corrects rejection of distractors (‘Correct rejection’, 3B), and accuracy of N-back task performance (‘Correct vs. Incorrect’, 3C). Only the significant ROIs that survived the FDR correction are shown. (**A**) 1—superior parietal lobule; 2—inferior parietal lobule (SMG posterior); 3—superior frontal gyrus. (**B**) 1—superior parietal lobule; 2—superior frontal gyrus (paracingulate gyrus); 3—inferior parietal lobule (SMG posterior). (**C**) 1—superior parietal lobule; 2—inferior parietal lobule (SMG posterior); 3—superior frontal gyrus (paracingulate gyrus). Coordinates are presented in Table 2.

**Figure 3 brainsci-11-00155-f003:**
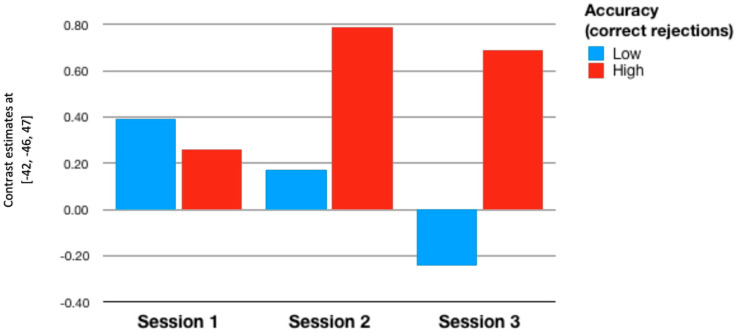
Contrast estimates in superior parietal lobule depending on the scanning session and accuracy in the N-back task. Contrast 1 (‘Hits’). Visualization has been prepared independently of the statistical computations, in which the accuracy variable was kept continuous.

**Figure 4 brainsci-11-00155-f004:**
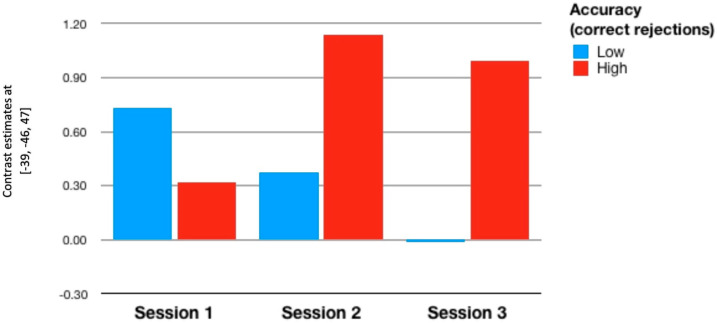
Contrast estimates in superior parietal lobule depending on the scanning session and accuracy in the N-back task. Contrast 3 (‘Correct vs. Incorrect’). Visualization has been prepared independently of the statistical computations, in which the accuracy variable was kept continuous.

**Table 1 brainsci-11-00155-t001:** Descriptive statistics concerning the N-back task. The number of hits (max = 16) and correct rejection of lures (max = 15) for two groups and three scanning sessions.

		Hits		Correct Rejections	
	Group	Mean	SD	Mean	SD
Session 1	Control	11.42	3.11	12.82	3.04
	Training	10.78	2.20	12.67	2.79
	All	11.10	2.69	12.75	2.89
Session 2	Control	13.97	1.60	14.25	1.22
	Training	12.86	2.78	13.75	1.61
	All	13.41	2.31	14.00	1.43
Session 3	Control	13.87	1.75	14.48	0.63
	Training	12.41	3.05	14.16	1.15
	All	13.13	2.57	14.32	0.93

**Table 2 brainsci-11-00155-t002:** Summary of analysis results corresponding to the significant clusters of activation for N-back task. The coordinates of the overlapping ROIs are also included, together with their anatomical label and corresponding Brodmann’s area number.

ROI Id	*X*	*Y*	*Z*	Hemisphere	*p* Value (FDR Corrected)	Name WFU	Anatomical Label
1	−42	−46	47	Left	*p* = 0.07	BA.40	Superior Parietal Lobule
2	−39	−52	44	Left	*p* = 0.011	BA.40	Inferior Parietal Lobule (SMG posterior)
3	−6	20	47	Left	*p* = 0.007	BA.08	Superior Frontal Gyrus
1	−30	−55	47	Left	*p* = 0.044	BA.07	Superior Parietal Lobule
2	−6	8	56	Left	*p* = 0.001	BA.06	Superior Frontal Gyrus (paracingulate gyrus)
3	36	−49	41	Right	*p* = 0.001	BA.40	Inferior Parietal Lobule (SMG posterior)
1	−39	−46	47	Left	*p* = 0.001	BA.40	Superior Parietal Lobule
2	36	−52	44	Right	*p* = 0.102	BA.40	Inferior Parietal Lobule (SMG posterior)
3	−6	8	59	Left	*p* = 0.001	BA.06	Superior Frontal Gyrus (paracingulate gyrus)

**Table 3 brainsci-11-00155-t003:** Summary of coordinates and analysis results corresponding to the significant clusters of activation for Stop-signal task.

Coordinates	*p* Value	*p* Value	Anatomical
(*X*, *Y*, *Z*)	(Uncorr.)	(FDR-Corr.)	Labels
‘Stop vs. Go’			
(36, 26, −4)	*p* < 0.001	*p* = 0.986	Right Insula
(−3, −28, 62)	*p* < 0.001	*p* = 0.986	Left Paracentral Lobule
(−45, 26, −1)	*p* < 0.001	*p* = 0.986	Left Paracentral Lobule
(18, −64, 5)	*p* < 0.007	*p* = 0.986	Right Calcarine
‘Stop vs. Fail’			
(−15, 23, 47)	*p* < 0.01	*p* = 0.98	Left Superior Frontal G.
(45, −19, 50)	*p* < 0.005	*p* = 0.98	Right Postcentral Gyrus

**Table 4 brainsci-11-00155-t004:** Spearman’s correlation coefficients between brain activations in contrast 3 (‘Correct vs. Incorrect’) and two measures of accuracy in the N-back task: the number properly detected targets (Hit) and the number of correct rejections (Reject).

**All (*n* = 46)**						
	Hit 1	Reject 1	Hit 2	Reject 2	Hit 3	Reject 3
RoI 1	0.062	−0.17	0.202	0.082	0.07	0.452 **
	*p* = 0.683	*p* = 0.259	*p* = 0.177	*p* = 0.588	*p* = 0.646	*p* = 0.002
RoI 2	−0.053	−0.089	0.103	−0.065	−0.068	0.389 **
	*p* = 0.727	*p* = 0.557	*p* = 0.497	*p* = 0.669	*p* = 0.651	*p* = 0.008
RoI 3	0.095	−0.029	0.177	0.109	−0.168	0.12
	*p* = 0.532	*p* = 0.847	*p* = 0.238	*p* = 0.469	*p* = 0.264	*p* = 0.426
**Training (*n* = 23)**						
	Hit 1	Reject 1	Hit 2	Reject 2	Hit 3	Reject 3
RoI 1	0.141	−0.076	0.315	0.204	0.183	0.554 **
	*p* = 0.512	*p* = 0.731	*p* = 0.144	*p* = 0.351	*p* = 0.404	*p* = 0.006
RoI 2	−0.034	−0.164	0.248	0.088	0.05	0.491 *
	*p* = 0.879	*p* = 0.455	*p* = 0.254	*p* = 0.689	*p* = 0.822	*p* = 0.017
RoI 3	0.026	0.046	0.197	0.407	−0.108	0.1
	*p* = 0.907	*p* = 0.835	*p* = 0.367	*p* = 0.054	*p* = 0.624	*p* = 0.651
**Control (*n* = 23)**						
	Hit 1	Reject 1	Hit 2	Reject 2	Hit 3	Reject 3
RoI 1	0.06	−0.227	0.09	−0.082	−0.163	0.377
	*p* = 0.784	*p* = 0.297	*p* = 0.682	*p* = 0.708	*p* = 0.459	*p* = 0.076
RoI 2	−0.098	−0.086	−0.013	−0.179	−0.226	0.371
	*p* = 0.656	*p* = 0.698	*p* = 0.953	*p* = 0.415	*p* = 0.219	*p* = 0.082
RoI 3	0.14	−0.106	0.075	−0.157	-0.404	0.113
	*p* = 0.525	*p* = 0.629	*p* = 0.732	*p* = 0.474	*p* = 0.056	*p* = 0.608

Note: RoI—region of interest; RoI 1—left superior parietal lobule; RoI 2—right supramarginal gyrus posterior; RoI 3—left paracingulate gyrus; Hit 1—the number of hits in session 1, etc.; Reject 1—the number of correct rejections in session 1, etc.; *p* values are two-tailed; ** *p* < 0.01, * *p* < 0.05.

## Data Availability

Data available at request.

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
