# Peer review of "The Effects of Working Memory Training on Brain Activity"

_brainsci, 2021, doi:10.3390/brainsci11020155_

Round 1

Reviewer 1 Report

The authors investigate in a very complex design whether 2 weeks of WM training would lead to changes in brain regions that are used during a WM task and test for transfer effects (inhibition, fluid intelligence). I congratulate the authors on their attempt to give an answer to this relevant question, but I do have some strong concerns specifically with respect to the design and the fMRI analyses.

  1. Could you specify the difference between the training and control conditions in the abstract?
  2. I suggest that you mention already in the abstract that no difference in accuracy was expected during the scan sessions due to the fixed and very easy N-back task (2-back) and maybe mention why you chose to do that (to investigate efficiency)
  3. The conclusion in the abstract is a bit confusing and I am not sure I understand what exactly you mean by this.
  4. I am missing some references (e.g. lines 49-53). Please make sure you are not missing any references.
  5. It may be important to discuss Poldrack’s paper from 2015 on the concept of efficiency here (Is “efficiency” a useful concept in cognitive Neuroscience?)
  6. Why was the SST also done repeatedly and not just once at the end to investigate clean transfer effects? Because participants now also trained the SST to some extent
  7. Do you mean “make performance rather high” here? (lines 230-231)
  8. Why would the SST otherwise not be suitable for fMRI? (line 247)
  9. Did you base the choice of 0 and 200ms intervals in the SST based on previous studies?
  10. FDR instead of FEW correction may lead to a higher false positive rate. Can you provide a short explanation why you chose FDR here?
  11. My biggest concern is the possibility of double dipping in the fMRI analyses
  • If any knowledge of the results leaks into the choice of small volumes the resulting statistics will be biased, inflating Type I error - see data-driven ROI selection without inflating Type I error rate (Brooks, Zoumpoulaki and Bouwman, Psychophysiology, 2016)
  • Can you specify in the methods exactly which contrasts were used to define the ROIs and which contrasts were used to test for the effects of interest? The results statistics should be independent of the selection criteria under the null hypothesis. (see Kriegeskorte, Nature Neuroscience, 2009). Please provide an explanation for all analyses why your data-driven ROI selection is not biased.
  • The same goes for the correlation with task performance. If you look at an ROI define by the contrast hits vs. baseline, this contrast will show selective activation for better performance. Do you think it is problematic if you add accuracy as a covariate then?
  1. My other concern is the choice of a 2-back task in the scan sessions
  • During the first stage of training sessions, trained subjects already have an average N level of 3.95. We can assume a 2-back task is very easy for both groups. If the N-back task is indeed easy for both groups, as predicted by you I would not predict a difference in performance. However, this does not fit why apparently performance can still improve between sessions as you show. Why should the training group then not be better than the control group (if there apparently is still room for improvement)?
  1. Could you write in the table 1 description which task results are shown here?
  2. Could you please provide a definition and reasoning for when you define results as “marginally significant” in the methods (for which p-vales and for which effect sizes)?

Author Response

  1. Could you specify the difference between the training and control conditions in the abstract? Done.
  2. I suggest that you mention already in the abstract that no difference in accuracy was expected during the scan sessions due to the fixed and very easy N-back task (2-back) and maybe mention why you chose to do that (to investigate efficiency). Done.
  3. The conclusion in the abstract is a bit confusing and I am not sure I understand what exactly you mean by this.  Thanks to your comment we realized  that the conclusion in the abstract was really awful. It has been changed.
  4. I am missing some references (e.g. lines 49-53). Please make sure you are not missing any references. Three references have been added: Kane & Engle (2001), Chuderski (2013), Gathercole et al. (2016).
  5. It may be important to discuss Poldrack’s paper from 2015 on the concept of efficiency here (Is “efficiency” a useful concept in cognitive Neuroscience?) This paper has been included into references and its content has been used in the discussion section.
  6. Why was the SST also done repeatedly and not just once at the end to investigate clean transfer effects? Because participants now also trained the SST to some extent. Yes, they also trained due the repetitions of consecutive testing phases. However, they did not train in between the phases. We expected transfer effects from the training proper, which did not occur. It was possible to use SST only in the final testing phase but such a decision might be susceptible to criticism as well. For instance, lack of expected transfer effects might be attributed to the fact that the final phase of n-back scanning is repeated twice whereas the final stage of SST scanning takes places for the first time, so the conditions are not comparable.
  7. Do you mean “make performance rather high” here? (lines 230-231) Yes, thank you very much for your watchfulness. It has been corrected.
  8. Why would the SST otherwise not be suitable for fMRI?  Indeed, the word "suitable" was not the best choice. We replaced it with the words "more convenient". It is of course possible to use the classic version of this task in the fMRI investigation (e.g., Chao et al., 2009). However, such a decision would cause many technical problems. This task needs that delay time was adjusted individually to every participant depending on his/her latest response. Although it is fairly possible at the software level, it produces unequal number of comparable events (e.g., delay 150 ms or delay 270 ms, and so on) for each individual. Some events might occur only in the case of selected participants, since others might show different patterns of responding. In consequence, such subtleties would have to be ignored while analyzing the brain contrasts. For instance, Chao et al. (2009) compared just four possibilities: success GO trials, success STOP trials, failure GO and failure STOP, although they made their participants to run the whole procedure within the scanner. Since the Stop Signal task was an auxiliary procedure in our study, not the main one as n-back, we decided to simplify it.
  9. Did you base the choice of 0 and 200ms intervals in the SST based on previous studies? To some extent, yes. In our own study (Nęcka et al., 2018, Frontiers in Psychology) we found that ∼400 ms of the mean value of stop signal delay ensured 50% mean accuracy. In an fMRI study (Chao et al., 2009, BMC Neuroscience) such a delay value was a bit shorter (∼350 ms). Since we did not run the regular extensive Stop Signal procedure, in which SSRT is usually computed as the difference between reaction time in correct trials and (constantly adjusting) stop signal delay that ensures 50% accuracy (Verbruggen, F., Logan, G. D., and Stevens, M. A., 2008), we had to decide on such delay values that would allow contrasting relatively easy and relatively more demanding versions of the task. So, 0 ms of stop signal delay seemed a natural decision for the easy version, and 200 ms was a disputable decision for the more difficult version (but not too difficult at the same time). Since 400 ms ensures 50% accuracy, the value two times shorter than that (i.e., 200 ms) seemed suitable to ensure the accuracy indices somewhere in between 50% and 100%. It must be underscored that 50% accuracy might be too low to exclude the possibility of guessing. As a matter of fact, we obtained 77% accuracy level in the hard version, which is a suitable level for analyzing individual differences.
  10. FDR instead of FEW correction may lead to a higher false positive rate. Can you provide a short explanation why you chose FDR here? Following the example of Kessler, Angstadt and Sripada (2017) we decided to focus on the false discovery rate (FDR), which is a more natural target for multiple testing control. According to Nichols (2016): "A researcher is naturally more concerned with the proportion of reported clusters that are false positives (FDR) than whether any are false positives (FWE)."

    Kessler, D., Angstadt, M., & Sripada, C. S. (2017). Reevaluating “cluster failure” in fMRI using nonparametric control of the false discovery rate. Proceedings of the National Academy of Sciences114(17), E3372-E3373.

    Nichols T.E., (2016). Bibliometrics of cluster inference. Available at blogs.warwick.ac.uk/nichols/entry/bibliometrics_of_cluster/.

  11. My biggest concern is the possibility of double dipping in the fMRI analyses
  • If any knowledge of the results leaks into the choice of small volumes the resulting statistics will be biased, inflating Type I error - see data-driven ROI selection without inflating Type I error rate (Brooks, Zoumpoulaki and Bouwman, Psychophysiology, 2016)
  • Can you specify in the methods exactly which contrasts were used to define the ROIs and which contrasts were used to test for the effects of interest? The results statistics should be independent of the selection criteria under the null hypothesis. (see Kriegeskorte, Nature Neuroscience, 2009). Please provide an explanation for all analyses why your data-driven ROI selection is not biased.
  • The same goes for the correlation with task performance. If you look at an ROI define by the contrast hits vs. baseline, this contrast will show selective activation for better performance. Do you think it is problematic if you add accuracy as a covariate then?

    The paper you are citing is a nice review on how to identify ERP (from EEG) ROI’s using a data driven method – but it seems not relevant to fMRI.

    None the less the issue of inflating Type I error in fMRI data-driven defined ROI’s is related to a standard and ill-advised approach of selecting ROI’s based on fMRI peak of activations. I.e. creating spheres around hypothesis significant peaks. In that case, it is double-dipping, and it inflates the Type 1 error. In contrast, what we did is the use of some global contrast to identify activated areas then, use the watershed transform to parcellate these areas to massive data-driven regions of interest. These ROIs were used to investigate learning effects at the ROI level. There was no need for the watershed as the activation maps are independent of each other anyhow. Instead of taking the peaks of activations, we used mean activity across large ROI’s.

    Furthermore, we defined the ROI’s in a way that was blinded to the learning effects. So the ROIs are isolating areas where we are sure there are differences between the control and treated groups regardless of the learning stage. Then we were explicitly asking the question of whether learning effects exist within these ROIs. So we increased the power (by reducing the multiple comparisons from tens of thousands to three) and did our best to help the test find some significant results – the fact that there isn’t any is the surprising thing, not the vice versa.

    So we believe we are not double dipping here.

  1. My other concern is the choice of a 2-back task in the scan sessions
  • During the first stage of training sessions, trained subjects already have an average N level of 3.95. We can assume a 2-back task is very easy for both groups. If the N-back task is indeed easy for both groups, as predicted by you I would not predict a difference in performance. However, this does not fit why apparently performance can still improve between sessions as you show. Why should the training group then not be better than the control group (if there apparently is still room for improvement)? The two groups did not differ significantly in n-back accuracy within the scanner, although the performance was not 100% accurate (see Table 1). There was a small increase in accuracy in the second phase, which shows that, indeed, there was still room for improvement, but this increment occurred in both groups. So, it looks like some non-specific effect of adaptation to the unusual circumstances of being tested within the scanner. Lack of such increment in the third phase may justify such an explanation. Outside scanner (that is, during training proper), the experimental group were constantly improving their performance but in this case the n number was increasing with progress in the adaptive way.
  1. Could you write in the table 1 description which task results are shown here? Done, its' n-back.
  2. Could you please provide a definition and reasoning for when you define results as “marginally significant” in the methods (for which p-vales and for which effect sizes)? By "marginally significant" we mean the effects where p value exceeds the conventional threshold of 0.05 but it is not greater than 0.10. We believed that, since p < 0.05 is just a convention, it may be interesting to look at effects where the probability of mistaken H0 rejection, when it is true, is a bit higher than 0.05, but still low. On the other hand, we do not call these effects "significant" because of the convention. Anyway, we removed this wording and replaced it with "a trend", so as to satisfy another reviewer.

Reviewer 2 Report

The current manuscript reports on a study that examined neuropsychological measures of working memory and neuro imaging among a group of individuals who received working memory training and a control who didn't over three waves of data collection. The manuscript is strengthened by the use of longitudinal data that allows for the examination of within-person changes, as well as an experimental group design. Additionally, the researcher's use of multiple types of outcome data provides a more comprehensive idea of what effects the dependent variables have on the constructs. This study has the potential to contribute significantly to the field, as the malleability of cognitive domains has implications for clinical and non-clinical research.  However, before being able to provide a recommendation for this article to be published, there are several items that need to be addressed which are outlined below (along with the strengths). Specifically, it is difficult to understand how some models were run based on the very little description of their analytical approach. It is difficult to interpret their results without further clarity. Additionally, there are problems with interpretation of results and a few gaps in theoretical modeling that need to be addressed. Without consideration of these items, the validity of the findings is less salient, and the likelihood of reproducibility is less clear.

  1. The authors did an excellent job introducing working memory and reviewing some of the work on the ability to improve WM through training. However, there doesn't seem to be much discussion on the actual WM training itself. As a reader, I would like to know, what does training working memory look like? Across the studies that have been conducted, is the processes of training working memory fairly standardized, and if so, what are the mechanisms that are theorized to improve WM through the "training"? This carries important implications for evaluating the current study's training method and whether or not their method was congruent with standard WM training practices. In regard to the neural mechanisms that explain differences in working memory, the authors do provide information on how improved neural efficiency by reducing activity from other parts of the brain by explain individual differences in WM , but then also suggest that working memory training strengthens neural connections. It would be helpful for the authors to improve their organization for what they think working memory training does on a neurological level that improves working memory (neural mechanism of change) and then directly state how that reflects in the hypothesis and their rationale for their models.
  2. With using the stop-signal task as an additional neuropsychological measure, inhibition is introduced as an executive functioning measures being examined. I understand the rationale for the use of the stop-signal task to examine the transfer effect, but why this task? There is definitely theoretical justification to focus on the transfer of WM to inhibition over choosing to look at other EF domains, but this isn't stated in the review at all, which doesn't provide the reader with the rationale for use of the stop-signal task. The authors need to explain why they used this task for inhibition to test for the transfer effect.
  3. Across the results section, the author's do not present their Null statistic results, but only the marginally significant and significant. Across all hypothesis testing, please provide all statistics for your results.
  4. Overall, there are major problems with trying to use a continuous moderator with such a small sample for the individual differences analyses in the way that it is currently being analyzed. Here, you have conceptualized the model as the covariates being a moderator of the relation between time and activation of brain regions. Thus, when you look at those who are "high" vs "low" in accuracy (or other measures) and using standard moderation probing techniques of defining high and low as +/- 1 SD (which needs to be mentioned and cited) you are looking the contrast between about 6 individuals at the high end and 6 individuals at the low end, which is not enough! Conceptually, you do not have the sample size to interpret that moderation of a continuous value at the person level. Overall, the bar graphs are very misleading as the reader does not know that they are seeing a comparison of such a small amount of people at the extreme ends. Furthermore, statistical probing of these interaction effects is necessary, It is one thing to say that the interaction is significant, but how is it significant? This would be difficult to do with the current method of probing across the moderator as you would have to create contrast coefficient for the High and low. What you could do, is examine time as a moderator and then explore how the slope between your covariates and brain activation vary across time. Here you would just have three lines, one for each time (and it would be preferred if you did a scatter plot to confirm that your results are not simply due to an outlier). With this suggested interpretation of the interaction (time moderating the relation), you could just look at the differences in slopes across time.

Additionally, there are other problems with these analyses and how they are written.

  1. Are the main effects of these models that include the N-back and stop-signal similar to examining the activation of these regions in the previous analyses? you already indicated that that activation left superior parietal lobule was marginally related to N-back hit rate (after FDR correction), but then you then report the relation between this region and N-back accuracy again in these analyses. How is this conceptually different? Hit rate and accuracy are going to be extremely similar. Additionally, these main effects cannot be interpreted as the interaction is significant. It is repeatedly mention, in the results and the discussion, main effects are interpreted as "more activation of this region XX is associated with XX" when the relation is actually conditional on the time point in many of the results. based on figure three, if someone where to test that themselves, they would have null findings at an initial time point, as the differences only appear to occur at time points 2 and 3. Thus, the researchers need to remove main effect interpretations and discussing of them when the interaction of time is significant.
  2. It needs to be clearly stated what data is being used as the time covariates and moderators, particularly for the N-back and stop-signal scores, as these could be time varying. I am assuming that the moderator of individual differences for all of these analyses is the initial ability across the covariates at baseline, as you also look at general intelligence, which was only measured once, but it is really difficult to confirm that. Did you use general intelligence as time invariant moderator and then the Nback and Stop-signal vary with time? In general, more detail for these computations is necessary.
  3. why did you not correct for multiple hypothesis testing with these analyses like you did with the others? if, in fact these are all "exploratory", please state this more than once in the discussion and be very clear and transparent when discussing these findings. I understand that it is worth running secondary analyses in addition to primary aims that you may not have the sample size to correct for type I error inflation, but please be clear that these are preliminary findings.
  4. A major problem with the interpretation of the null effects of the model is that the authors state that they did not find effects for certain interactions between the treatment group when there was no power analysis or consideration of what sample size would be required to detect an effect. This is also a problem related to the lack of null finding statistics. Were patterns trending in the hypothesized direction? Were the effects too small to detect with the given sample size? All of these are crucial questions to ask when determining whether a effect was not detected due to the hypotheses being wrong, or if research design prevented you from detecting the effect. This is also important as many of these effects appear to be trending in a pattern. For example, in table 4 where the authors provide many of the spearman correlations, a number of the non-significant effect are congruent with a medium or even large effect.
  5. Although mentioned previously, sample size for this study is not even considered throughout. There are many replication problems in neuro imaging studies related to small sample sizes. The problems that this has created have already been outlined, but addressing the lack of power analysis in a general statement is necessary. Accordingly, any mention of "rejecting the hypothesis" in the manuscript should be changed, because, although you failed to reject the null hypothesis, decisions on whether your alternative hypothesis is based on beta, which is influenced by sample size, and effect size, which are not approached within this paper.

Minor revisions

  1. The introduction is quite extensive, which is reflective of the author's comprehensive review, but it can be a lot of information to synthesize for the reader. Adding subheadings to the introduction and providing sentences that synthesize topics within would be helpful with breaking the large introduction into specific topics.
  2. line 51 and 52 of page 2 references "multiple studies", but there is no citation here for the reference being made about these studies. Please provide the citations for this claim.
  3. similar to the previous comment, line 57 and 58 reference "recent studies" but it is not clear what citation the author is referring to as there are no immediate citations in that sentence.
  4. the wording on line 457 is not correct statistically speaking, as we do not reject alternative hypothesis, instead we fail to reject the null. Decisions on whether you can actually clearly state that the alt is reject, is actually based on power.
  5. Please state if the sample that was used was the complete sample or if it was from the removal of missing data. If there was absolutely no missing data from the sample and the sample in the study is based off of every single participant who was enrolled, please state this as this was a longitudinal study that required multiple data collection and drop out is common.
  6. Table 4 should include standard error and P value of every analyses.

Author Response

1. There is nothing like the standard WM training in the literature. Following the study by Suzanne Jaeggi and co-workers (Jaeggi et al., 20080), the dual n-back procedure is sometimes applied, where participants have to detect repetitions of both acoustic and visual stimuli that appear simultaneously. This variant of n-back is quite difficult, so we decided not to use it in the fMRI settings. Had we known that Salminen et al. (2020) used dual n-back in the scanner, we would probably decide on such a solution. However, our study was planned and carried out before their publication. Dual n-back is sometimes used in the WM trainings without any brain scanning. In fMRI studies, single n-back paradigms are usually applied. Sometimes it is a spatial version of this task, like in Buschkuehl et al. (2014). The design we adopted was similar to other studies of this kind, including the studies recently published (e.g., Miró-Padilla, A., Bueichekú, E. & Ávila, C. Locating neural transfer effects of n-back training on the central executive: a longitudinal fMRI study. Sci Rep 10, 5226 (2020). https://doi.org/10.1038/s41598-020-62067-y). Moreover, we tried out this version of n-back with non-verbal stimuli in some non-imaging studies published by ourselves (e.g., Chuderski & Nęcka, 2012). 

2. The rationale for using the stop-signal task has been added.

3. The null statistics results should be added. We did not include them yet because of the deadline but this is a job that should be done in a few days.

4. The moderator variables were kept continuous during all statistical computations. We split the sample in the median point just for graphical illustration, which is clearly stated in the figures 3 and 4 captions. Maybe I don't understand this issue but surely we did not compare as small number of participants as six. The suggestion that we should examine slopes representing relationships between the covariates and brain activations as they differ in three time points is very interesting and helpful. Due to deadline we could not include such visualization into the corrected manuscript but we will surely try to do such analyzes.

Points 3-4-5-5: I have problems in getting the point. Let me underscore that there were two indices of c=accuracy in n-back task: hit rate and accurate rejection of lures. We differentiate them in the manuscript, including figure captions. If they are somehow mistaken, I would be eager to correct.

We underscored that our findings are preliminary and that there are many caveats.

Table 4 is corrected so as to include all p values, even non-significant.

5. The sample size is an issue we discuss in the newly added fragment of the last paragraph before Conclusions. In general, many other neuroimaging studies on WM training are based on sample size around 50 persons (46 in our study), so we don't think that our study is somehow deviant concerning sample size. However, analyzes of individual differences may need more statistical power.

Minor revisions:

6. The introduction has been divided into three subsections with subtitles.

7. Lacking citations have been added.

8. Lacking citations have been added.

9. We have changed the wording, saying that the hypothesis "did not obtain empirical support" instead of it "should be rejected".

10. Forty-six out of 50 participants who were originally enrolled finished the study. Although the study was longitudinal, all the testing took only two weeks for every individual. What is more, the financial compensation was quite generous and paid after everything has been completed. Thanks to these factors we had only four drop-outs. This issue has been mentioned in the manuscript.

11. Table 4 has been changed, p values are included.

Reviewer 3 Report

Review

This investigation aims at illustrating the effect of working memory training and its impact on the neural network. Therefore, n-back tasks were used and a Stop-signal paradigm to find possible transfer effects.

Generally speaking, this is a very interesting paper. So, I have some minor aspects which should be addressed and fixed. Section 3 is most important. Here I suggest that you mention statistical procedures for the entire investigation in the opening so that the readers can follow your line of argumentation. In addition, a list of variable names or some lines which illustrate them should be included (see all my comments later). I listed the points including the line numbers for all sections below.

  1. Introduction

Lines 106-109   The last part of the sentence where it says “, although a meta-analysis [27] shows that the effect size of training does not depend on this factor” is difficult to read. If you were so kind and add some background information, this sentence may be easier to get.

Lines 145 and 146 – two times hypothesis ( leave it up to the authors to change this..)

Lines 179 -180 The sentence “We chose a simplified …. “ maybe you can include one short sentence which explains the Stop-signal paradigm here before you do so in the later part.

Lines 182 There were three training sessions… Is it possible to mention the different testing phases? Meaning middle training for instance Middle training (after 6 days) etc..(see also my comments later)

  1. Materials and Methods
  • Participants

As you said you have a ratio of 2:1 for male and female participants. Did you also check whether gender has an effect on matching and on WMC , Raven, …etc. ? Also, the matching procedure is something people are interested in. So, maybe you can include 2 more sentences about it.  

  • Study design

See comment above lines 182 could you specify what middle means. I also could not find it in 2.4.1 The training procedure.  I think this is important. Thus, I suggest that you add this information once in a section which suits you best!

        2.3.2 Stop-signal task

Lines 241-244 are difficult to read. Please revise.

Lines 273 first word “before”. - Before what takes place?

  1. Results

You run a number of different statistical analysis for good reasons. Therefore, I suggest that in the opening of the results section you should write a very short passage which includes the rational idea of your statistical analysis and why you performed this in the way you did.  In this section it may also be worth to include few sentences which define the variable names e.g. session etc.

Lines 322 -339

Anova repeated measures is fine but instead of subtracting numbers and perform t-tests you could have run a growth model where you do not have to correct for multiple testing (you don’t have to do one for now but it maybe a better approach for future research).  T-tests should be Bonferroni corrected if you run more than one. As I see, based on the values of your t-tests both remain significant after correction which is fine.

Lines 389 -390/ 393 / 421“marginally significant” – I would say they are non-significant. If you still wish to put emphasis on the contrasts, you could probably speak of a trend…

Lines 408 “contras” I think a “t” is missing

Line 428 “than” you mean “that”?

Line 480 “and the factor of session” I think you should omit “of”.

Lines 490-491 is difficult to read

Line 500 – “Anyway”  maybe you find a better word ? (also line 642)

Table 4 – this table should be improved in design. Also, include the number of participants in each group. I also think that you should not overestimate correlational analysis and make too strong claims, especially as correlations are rather instable when groups are small. So whenever you talk about correlations I would include wordings like “ this maybe illustrate” or “this seems to indicate that” to make your claims less strong (e.g. lines 541-549 and in the discussion)

Line 541: “these two parietal regions changed…” please specify them

Author Response

Lines 106-109   The last part of the sentence where it says “, although a meta-analysis [27] shows that the effect size of training does not depend on this factor” is difficult to read. If you were so kind and add some background information, this sentence may be easier to get. This sentence has been split into two sentences and rephrased in order to make it easier to get.

Lines 145 and 146 – two times hypothesis ( leave it up to the authors to change this..) We'd rather leave it as it is.

Lines 179 -180 The sentence “We chose a simplified …. “ maybe you can include one short sentence which explains the Stop-signal paradigm here before you do so in the later part. We added short explanation: "This task was introduced in order to check possible transfer effects. We chose a simplified version of the Stop-signal paradigm [48], which requires that an already activated behavioral tendency be contained in response to the signal of stop."

Lines 182 There were three training sessions… Is it possible to mention the different testing phases? Meaning middle training for instance Middle training (after 6 days) etc..(see also my comments later) You probably mean "scanning sessions". This fragment has been rephrased.

As you said you have a ratio of 2:1 for male and female participants. Did you also check whether gender has an effect on matching and on WMC , Raven, …etc. ? Also, the matching procedure is something people are interested in. So, maybe you can include 2 more sentences about it.  Gender had an effect on matching because we first matched volunteers in pairs for age, sex and level of education, and next we randomly assigned them to the training or control condition, as specified in the Participants section. We did not check for gender differences concerning WMC and Raven because this was not the primary topic of interest.

Lines 241-244 are difficult to read. Please revise. Revised.

Lines 273 first word “before”. - Before what takes place? The day before = yesterday, the day preceding the actual training session. It has been rephrased.

You run a number of different statistical analysis for good reasons. Therefore, I suggest that in the opening of the results section you should write a very short passage which includes the rational idea of your statistical analysis and why you performed this in the way you did.  In this section it may also be worth to include few sentences which define the variable names e.g. session etc. Thank you for this suggestion. We added a short paragraph at the beginning of Results section.

Anova repeated measures is fine but instead of subtracting numbers and perform t-tests you could have run a growth model where you do not have to correct for multiple testing (you don’t have to do one for now but it maybe a better approach for future research).  T-tests should be Bonferroni corrected if you run more than one. As I see, based on the values of your t-tests both remain significant after correction which is fine. Thank you for these comments.

Lines 389 -390/ 393 / 421“marginally significant” – I would say they are non-significant. If you still wish to put emphasis on the contrasts, you could probably speak of a trend… It has been changes for trend.

Lines 408 “contras” I think a “t” is missing Corrected, thank you.

Line 428 “than” you mean “that”? Corrected, thank you.

Line 480 “and the factor of session” I think you should omit “of”. Removed.

Lines 490-491 is difficult to read Revised. 

Line 500 – “Anyway”  maybe you find a better word ? (also line 642) Replaced with "Altogether" and "Still", respectively.

Table 4 – this table should be improved in design. Also, include the number of participants in each group. I also think that you should not overestimate correlational analysis and make too strong claims, especially as correlations are rather instable when groups are small. So whenever you talk about correlations I would include wordings like “ this maybe illustrate” or “this seems to indicate that” to make your claims less strong (e.g. lines 541-549 and in the discussion) Table 4 has been rewritten. Number of participants and p values are included. Conclusions and claims have been weakened in the discussion section, according to your suggestions. 

Line 541: “these two parietal regions changed…” please specify them Specified.

Round 2

Reviewer 1 Report

Dear authors

thank you very much for your detailed responses and explanations. I have no further concerns.